# Melkersson–Rosenthal Syndrome in Childhood: Report of Three Paediatric Cases and a Review of the Literature

**DOI:** 10.3390/ijerph16071289

**Published:** 2019-04-10

**Authors:** Salvatore Savasta, Alessandra Rossi, Thomas Foiadelli, Amelia Licari, Anna Maria Elena Perini, Giovanni Farello, Alberto Verrotti, Gian Luigi Marseglia

**Affiliations:** 1Pediatric Clinic Fondazione IRCCS Policlinico San Matteo–V.le Golgi, 19 Pavia, Italy; s.savasta@smatteo.pv.it (S.S.); alesross@hotmail.com (A.R.); t.foiadelli@smatteo.pv.it (T.F.); a.licari@smatteo.pv.it (A.L.); annamariaelena.perini01@universitadipavia.it (A.M.E.P.); gl.marseglia@smatteo.pv.it (G.L.M.); 2Pediatric Clinic–Department of Life, Health and Environmental Sciences–Piazzale Salvatore Tommasi 1, 67100 Coppito (AQ), Italy; giovanni.farello@cc.univaq.it; 3Pediatric Clinic–Biotechnological and Applied Clinical Sciences Via Vetoio (Coppito 2), 67100 Coppito (AQ), Italy

**Keywords:** Melkersson Rosenthal Syndrome, fissured tongue, peripheral facial palsy

## Abstract

Melkersson–Rosenthal syndrome (MRS) in children is a rare condition, clinically characterised by a triad of synchronous or metachronous symptoms: recurrent peripheral facial palsy, relapsing orofacial oedema, and a fissured tongue; the most recent review published on the topic has reported 30 published patients. The aetiology of this disease is still unclear. However, genetic factors, as well as alterations in immune functions, infections, and allergic reactions have been postulated. We report three children suffering from MRS and perform a literature review of paediatric cases. Taking into account that clinical and laboratoristical criteria for the diagnosis of MRS are lacking, this syndrome is probably underestimated, and we suggest increasing awareness of such a rare syndrome. Close multidisciplinary follow-up of these children with a team composed by paediatricians, neurologists, neuro-ophthalmologists, dermatologists, and otolaryngologists is crucial to guarantee exhaustive management and treatment success, while minimising relapses.

## 1. Introduction

Melkersson–Rosenthal syndrome (MRS) is a rare neuro-mucocutaneous condition of unknown aetiology, clinically characterised by a triad of synchronous or metachronous symptoms: recurrent peripheral facial palsy, relapsing orofacial oedema, and a fissured tongue [1,2]. However, only 8–25% of the cases show the complete triad, since the majority of the patients present with oligosymptomatic or monosymptomatic forms [1,3,4]. Painless recurrent orofacial oedema, also called Miescher MRS or cheilitis granulomatosa of Miescher, is the most common monosymptomatic presentation of MRS, and diagnosis is suggested by the identification of a non-caseating granuloma on a mucocutaneous biopsy of the subjects [4,5]. Oligosymptomatic or complete forms do not require additional bioptic investigation, since MRS is a clinical syndrome [6,7]. The aetiology of this disease is still unclear. However, genetic factors, alterations in immune functions, infections, and allergic reactions have been postulated [8,9,10,11]. In recent years, attention has raised about paediatric onset of the disease. We report three children brought to our clinic in whom we have posed the suspicion of MRS, and present a literature review of paediatric cases; the aim of this report is to increase awareness of such a rare syndrome.

## 2. Methods

All patients were evaluated in the Paediatric Clinic of IRCCS San Matteo Hospital Foundation, Pavia (Italy) from April 2014 to November 2018. Data collection was performed retrospectively from medical records. Detailed investigations are listed in Appendix A.

Literature review strategy was the following. We searched PubMed (Medline) through 31 August 2018 using the search string ((“Melkersson–Rosenthal” AND “Syndrome”) OR “Miescher”) OR (“Cheilitis granulomatosa” AND “Miescher”) OR (“facial palsy” OR (“lingua plicata” OR “furrowed tongue” OR “scrotal tongue”) OR (“orofacial oedema” OR “facial swelling” OR “lip swelling”)) AND (“paediatric cases” OR “children”).We also searched for the references of the related published articles. Paediatric cases of MRS were reviewed from detailed information when available, including sex, age at presentation, ethnicity, presence of facial paralysis with affected side and number of relapses, presence of orofacial oedema or lingua plicata, positive family history, comorbidities, and treatment. This information was compiled into a database using Microsoft Excel XP, and an extensive evaluation of the data was performed. Articles not available in the English language were included.

### Ethical Statement

We confirm that we have read the Journal’s position on issues involved in ethical publication and affirm that this report is consistent with those guidelines. All subjects gave their informed consent for inclusion before they participated in the study. Informed consent was obtained from the participant’s mother, and consent to use images was also included in the study. Informed consents will be available upon request to the corresponding author.

## 3. Case Report

### 2.1. Patient 1

A 14-year-11-month-old girl was referred to our Paediatric Emergency Department with dysgeusia and reduced mobility of the left side of the face, unable to close the left eye for one day before the admission. She had no remarkable recent medical history. When she was 11 years old she had a similar episode, with the inability to close the right eye and deviation of the labial commissure, which had completely disappeared after treatment with acyclovir and prednisone; at that time, parents denied any trauma, and House–Brackmann grade 4 facial palsy was diagnosed. A follow-up was planned for six months, and neurological sequelae or recurrences in that period were excluded. Her family history revealed that her father also suffered from recurrent peripheral facial nerve palsy. The physical examination showed right-sided deviation of the labial commissure, obliteration of the left nasolabial fold, incomplete closure of the left eye (Figure 1), swelling of the upper and lower lips, and a fissured tongue (Figure 2). There was no evidence of other cranial nerve involvement, and a detailed neurologic assessment did not reveal any other neurological deficits. The complete autoimmunity panel was performed, resulting in normal values except for ANA positivity (1:160). The recurrence of symptoms, results of laboratory tests, and instrumental assessments led to a suspicion of MRS. The patient was started on a tapering dose of prednisone for 5 weeks. She was given acyclovir until cerebrospinal fluid (CSF) analysis resulted negative for a viral load. Furthermore, the patient received intramuscular vitamin B-12 supplementation (500 mcg weekly for 5 weeks). At the four-month follow-up, there was no longer evidence of the facial palsy, and none of the symptoms have recurred during the last three years.

### 2.2. Patient 2

A girl aged seven years and eight months was referred to our observation because of left peripheral facial palsy, causing the inability to close the left eye and dropping of the corner of the mouth. A first peripheral facial nerve palsy occurred when she was three years and one month old, with complete regression after corticosteroid treatment. At the age of three years and nine months, she was diagnosed with pure red cell hypoplasia, manifested as severe anaemia (haemoglobin: 3.00 g/dL; red blood cells: 1,000,000/mm^3^) with an extreme lack of erythroid precursors in the bone marrow, but high growth of them in culture, probably caused by anti-EPO antibodies. The detection of anti-EPO antibodies, however, is not routinely performed in a clinical setting. The autoimmune hypothesis was postulated on empirical bases, since haemoglobin levels did not increase after recombinant human EPO administration, but normalized after corticosteroid therapy, and the addition of autologous serum to the erythroid precursor culture inhibited EPO growth. When she was 4 years and 4 months old, the patient presented with a second episode of left facial palsy, combined with the acute onset of a strength deficit on the left side of the body. Mingazzini I and II were positive for the left limbs. The imaging assessment showed a haemorrhagic stroke corresponding to the anterior portion of the right putamen and of the external capsule with perilesional oedema, involving the anterior limb of the internal capsule. Blood pressure measurements performed during the hospitalisation revealed high diastolic blood pressure values. These findings suggest a central rather than peripheral involvement of the facial nerve. Three weeks after their beginning, the symptoms had completely regressed. At the age of five years and six months, a third episode of left peripheral facial palsy occurred. Brain magnetic resonance imaging (MRI) was repeated, showing gliotic evolution of the previous haemorrhagic insult without new lesions. The patient was treated with corticosteroids, with a good regression of symptoms. On the last episode, the patient had initially visited a first level emergency room, where laboratory tests, as well as ophthalmologic, neurologic, and otoscopic examinations and a head computed tomography (CT) scan performed were normal. When admitted to our department, the neurologic examination showed complete peripheral left facial palsy (House–Brackmann grade V). Physical examination showed the presence of a furrowed tongue as a synchronous anomaly. No active herpetic mucosal and skin lesions were found. The patient was started on a tapering dose of prednisone for 30 days and vitamin B group supplementation was added. The clinical course was favourable. Three months after, at last follow up, neurological impairment had clearly improved. Facial palsy gradually resolved after the third week of treatment.

### 2.3. Patient 3

A nine-year-and-one-month-old girl was referred to our paediatric department with an acute right peripheral facial palsy, causing inability to close the right eye and periorbital pain (House–Brackmann grade IV). Symptoms had set in two weeks earlier, and since then she had undergone an otoscopic evaluation and a cranial MRI, with and without contrast; these tests had shown normal findings, except for a mild right facial nerve gadolinium enhancement. The child was started on oral prednisone, with little clinical benefit, and was therefore referred to our paediatric neurology unit. Her parents reported a previous episode of facial palsy concomitant with an acute otitis when she was 18 months old. A physical examination showed orofacial oedema involving the right cheek, while a neurological examination revealed right lagophtalmos and dropping of the right corner of the mouth, along with Bell’s sign positivity. Serological isoelectro focusing showed a previous infection with Cytomegalovirus and Epstein–Barr virus. The association between recurrent peripheral facial palsy and orofacial oedema, and the idiopathic nature of facial palsy itself suggested a diagnosis of MRS. The patient was started on a tapering dose of prednisone for 25 days; she was treated with acyclovir for 10 days, and received Vitamin B (daily oral administration for two months) and Vitamin D supplementation. At the one-month follow-up, the paralysis had been markedly reduced (House–Brackmann grade II).

## 4. Discussion

The reported incidence of MRS is between 0.2 and 80 in 100,000 per year. This incidence may be underestimated, since this syndrome is both mis- and under-diagnosed, and a mean diagnostic delay of nine years has been reported by several authors [2,6,7]. The first clinical manifestations of MRS typically occur in young adults between the second and third decade of life, with no racial predilection, although several oligosymptomatic cases can be diagnosed later in life [1]. MRS is rare in childhood, and the most recent review published on the topic has reported a maximum of 30 published case reports from the beginning of the 20th century [10]. Our analysis revealed that the number of children affected by this syndrome is much wider (Table 1).

As a matter of fact, the first symptoms occurred before the age of 18 years in 116 cases, but only 67 patients (57.8%) were diagnosed with MRS before this age [2,3,4,5,10,12,13,14,15,16,17,18,19,20,21,22,23,24,25,26,27,28,29,30,31,32,33,34,35,36,37,38,39]. Two of the three cases we have reported have been diagnosed under the age of 10; interestingly, literature describes only 18 cases in which diagnosis was made that early [10,14,15,16,17,18,27,31,34,35,37,39]. As highlighted in Figure 3, MRS in children is mostly diagnosed at between 7 and 12 years of age [5,10,14,16,17,18,19,22,24,27,29,34,35,36,37,39]. The youngest patient, described by Ehmann et al., was 22 months old [14]. The bar chart referring to the “Age distribution at first episode” shows that the onset of the disease covers a wider age interval, between 4 and 15 years (Figure 4) [2,3,5,6,7,12,13,16,17,19,22,23,24,25,26,29,30,31,33,35,36,37,38,39].

MRS seems to be more frequent among females [27,40]. So far, the cases reported in literature have highlighted a slight female preponderance before 18 years of age (44 males, 55 females (55.6%); see Table 1) [2,3,4,5,10,12,13,16,17,18,19,22,24,26,30,31,33,34,35,37,38,39]. Some authors have speculated that sex hormones may be important triggers or predisposing cofactors for episodes of orofacial oedema or lower motor neuron facial nerve paralysis [2]. Moreover, although the aetiology of MRS remains unknown, some authors suppose that the triad of symptoms could be caused by exposure to autoantibodies, which could partly explain this epidemiological data [10]. As a matter of fact, women build stronger immune responses against the non-self, but also against self-antigens, compared to men [41]. Lee et al. and Scagliusi et al. have documented the possible association between MRS and autoimmune disorders, describing, respectively, a female child and a male adult with Hashimoto thyroiditis [10,42]. Patient 2 was diagnosed with an autoimmune anaemia causing pure red-cell hypoplasia, 8 months after the first presentation of MRS. More studies are needed to determine whether the co-occurrence of MRS and autoimmune diseases is a coincidental event, or if it may be part of a same pathogenic mechanism.

Infectious agents like *Mycobacterium tuberculosis* and *paratubercolosis*, *Borrelia burgdorferi*, *Saccharomyces cervisiae*, and *Candida albicans*, as well as allergies to food and food additives have been considered as aetiological factors [10]. Genetic factors have also been postulated, since many authors have reported familial occurrence of the disease [7,19,38]. Lygidakis and colleagues described a male-to-male vertical transmission with autosomal dominant inheritance pattern in a family with seven affected members in four generations [19]. Xu et al. employed exome sequencing to determine potential mutations for MRS: mutations of FATP1, a fatty acid transport protein involved in fatty acid metabolism, has been identified as a causal gene for MRS in a Chinese family [38]. When familial history was investigated, another member of the family was found to be affected for 12 out of 30 children (40%) [2,4,5,6,7,10,13,15,16,19,23,25,28,33,35,36,37,38]. Only Patient 1 had a positive family history for signs and symptoms of MRS, her father being affected by recurrent peripheral facial palsy. A prospective investigation by Sun and colleagues showed that familial history for recurrent facial palsy was significantly higher in MRS (31.3%) than in Bell’s palsy (6.5%) [43]. However, genetic investigations have not yet identified single causative genes, and there is both clinical and genetic heterogeneity in MRS patients [44].

Beyond the cases with the complete clinical triad, patients can present with mono- or oligo-symptomatic forms. This clinical heterogeneity is reflected by the gross and unstandardized diagnostic criteria, whose requirements are the presence of at least two of the characterizing symptoms, or in the presence of isolated oro-facial edema, the identification of a non-caseating granuloma on a muco-cutaneous biopsy of the subject. Approximately one-third (38/116; 32.8%) of the patients in our literature review of MRS paediatric cases presented with the complete clinical triad [2,3,4,7,10,13,14,16,17,18,20,21,26,28,30,31,33,34,35,36,37,38,39]. As reported in adults, orofacial oedema is the most frequent symptom of MRS, with 63 of 116 children affected (54.3%; Table 1) [2,3,4,5,6,7,10,12,13,14,15,16,17,18,19,20,21,22,24,25,26,27,28,29,30,31,33,34,35,36,37,38,39]. Oedema is acute, non-pitting, and painless [24,29,35,36]. It is mostly confined to the lips, the upper one being more frequently affected, but it can also affect ipsilateral buccal mucosa, the nose, the eyelids, genitalia, or it can show a wider involvement of the face of the patients [4,12,17,18,20,22,24,26,36]. Oedema often persists for weeks and tends to recur [17,22]. Differential diagnosis with angioedema is based on the duration of the symptoms, on antihistaminic response, and on the natural history of the tissues involved: MRS oedema lasts longer, does not respond to histaminic therapy, and can lead to tissue fibrosis [7]. In our cases, hereditary angioedema was excluded by normal C1-esterase inhibitor levels. Lingua plicata is considered a developmental malformation, and it is estimated to affect from 0.5% to 5% of the general population [7]. This feature is reported in 56/116 (48.3%) cases from our review (Table 1) [3,4,6,7,10,13,14,16,17,18,20,21,22,24,25,26,29,30,31,33,34,35,36,37,38,39]. The percentage of adults with MRS showing this anomaly seems to be greater (50–70%), suggesting that this manifestation is less common (or less eye-catching) during childhood and may develop during the course of the syndrome, as Saini et al. describe for a 8-years-old girl [10,17,34]. Facial nerve paralysis occurs in 30–50% of the subjects with MRS [23,45]. Facial palsy can affect one or both sides, partially or completely [2]. In our review, facial palsy affected 61/116 (52.6%) paediatric cases with onset under the age of 18 (Table 1) [2,3,4,6,7,10,12,13,14,15,16,17,18,19,20,21,22,25,26,28,30,31,33,34,35,36,37,38,39]. Of the 41 cases in which side involvement was specified, 16 (39.5%) had recurrent homolateral facial palsy (eight left [4,10,16,17,26,28,31,35] and eight right [2,3,10,16,36,38,39]), 24 (58.5%) had alternating unilateral palsy [2,6,7,12,13,15,17,18,19,20,21,26,33,34,37], and one last patient (2.4%) experienced recurrent synchronous bilateral palsy [18]. Rivera-Serrano et al. found similar results in adults: 66.7% of subjects had alternating unilateral paralysis, and only a single case had a simultaneous bilateral facial paralysis [2]. Bell’s palsy recurs in 8.2% of cases, while in MRS facial palsy recurs in the majority of the subjects [17,46]. Relapsing episodes tend to last longer, with poorer recovery, with development of progressive fibrosis of the interested tissues or facial atrophy and masticatory muscles weakness [23]. As Wang et al. demonstrated, the prognosis of recurrent facial palsy in MRS is much worse than of recurrent Bell’s palsy [46]. Associated symptoms have been described in children with MRS, such as migraine, hearing loss and tinnitus, or dysgeusia (e.g., in Patient 1) [2,6,17]. At least some of these symptoms could be related to the involvement of additional cranial nerves. Studies have shown that trigeminal, hypoglossal, glossopharyngeal, auditory, and olfactory nerves may be involved in MRS [47,48,49]. Tinnitus could be a consequence of vestibulocochlear nerve lesion, although hyperacusia due to alteration in the stapedial reflex can sometimes be misinterpreted. Similarly, periorbital pain (complained about by Patient 3) and migraines can be a consequence of a trigeminal nerve dysfunction. Combined palsy of both the trigeminal and facial nerves may be clarified by taking into account the anatomical connections between them, which involve the three divisions of the trigeminal nerve [50].

Given its diagnostic challenges and rareness, MRS remains a diagnosis of exclusion. The absence of specific biomarkers for MRS implies that multiple laboratory and imaging tests are needed to assess the final diagnosis. Specifically, further investigations are recommended when no clinical improvement of a peripheral facial palsy has occurred after three weeks of therapy [51]. MRS in children must be differentiated from congenital conditions occurring with facial palsy, such as Mobius syndrome [31,52], Goldenhar syndrome [53], congenital pseudobulbar palsy, and Arnold–Chiari syndrome, which are typically non-progressive and associated to additional malformations and cranial nerve palsies [54]. Delivery trauma is a common cause of congenital facial palsy, especially in cases of prematurity, macrosomia, use of forceps, and caesarean section [51]. CSF analysis and serologies are commonly performed to exclude neural infections that can cause mono or bilateral facial nerve palsy, such as herpes simplex, varicella zoster, Epstein–Barr virus, Cytomegalovirus, tuberculosis, *Treponema pallidum*, and *Borrelia burgdorferi*. The reactivation of herpes varicella-zoster may be responsible for Ramsay Hunt syndrome. In this case, the association between facial paralysis and vesicular lesions of the auricular concha and the external auditory canal is highly suspicious. Lyme disease is a common cause of acute facial paralysis in children living in endemic areas [55]. Generally, infections need specific treatment to avoid long term sequelae. Interestingly, some authors suggest that even in the presence of active viral infections, MRS could be triggered by an abnormal immunological response to non-self-antigens, similarly to other autoimmune diseases involving the central nervous system (CNS) [8,9,10,11]. Head CTs and cerebral MRI are useful to exclude other conditions determining facial palsy, such as trauma (e.g., temporal bone fracture in younger children) or neoplasms, especially when additional, potentially confusing symptoms and risk factors are present. An otorhinolaryngologic evaluation allows the exclusion of acute and chronic otitis media or mastoiditis, which are very common causes of paediatric peripheral facial nerve paralysis [56]. Finally, granulomatous diseases can mimic MRS. The presence of non-necrotizing granulomas on the mucocutaneous biopsy of a patient with isolated oro-facial swelling suggests the diagnosis of Miescher syndrome. However, sarcoidosis and Crohn disease can share identical clinical and bioptic findings, and indeed the relationship between these entities has not yet been clarified [57]. Before any invasive test is performed, however, clinicians should rule out common oro-facial swelling aetiologies, which include allergic or hereditary angioedema as a major differential diagnosis, but also rosacea, foreign body reactions, tuberculosis, and Wegener’s granulomatosis [58]. Lingua plicata is usually considered a benign anatomical variant, and its prevalence in the general population has been reported in up to 10%. However, acquired conditions such as vitamin B deficiency may mimic clinical presentation [59,60].

Treatment guidelines for MRS are lacking. Although a spontaneous recovery is possible, currently the initial therapeutic approach is based on intralesional or systemic corticosteroids administration. In particular, a short course of systemic corticosteroids (i.e., prednisolone1-1,5 mg/kg/die), tapered over 3–6 weeks, depending on the severity of the manifestations, improves symptoms in 50–80% of the patients, with a recurrence rate decrease of 60–75% [10,17,27]. In addition, Stein et al. suggest the combined use of steroids with minocycline in children with MRS [22]. As HSV is a rare but potentially devastating cause of facial palsy, it is common practice to add antiviral agents (acyclovir) to steroid therapy, at least until exclusion of a viral etiology [34,37,39]. In a recent letter to the editor, Fantacci et al. proposed the use of intravenous immunoglobulins in a eight-year-old girl with MRS, suggesting a potential new treatment option in patients with unresponsive MRS [39]. When medical treatment fails, facial nerve decompression is effective in preventing further recurrence [61]. One of the limits of most retrospective studies is that patients received two or more therapies at the same time, limiting the efficacy assessment of any single management independently.

## 5. Conclusions

MRS in children is rare, and only few paediatric cases have been diagnosed so far. The association between facial oedema and facial paralysis in a child with a fissured tongue should alert the physician. Furthermore, symptoms can be non-simultaneous; therefore, diagnosis often requires repeated observations and a long follow-up. Close multidisciplinary evaluations of these children with a team composed by paediatrics, neurologists, neuro-ophthalmologists, dermatologists, and otolaryngologists is crucial to guarantee an exhaustive management and treatment success, minimising relapses.

## Figures and Tables

**Figure 1 ijerph-16-01289-f001:**
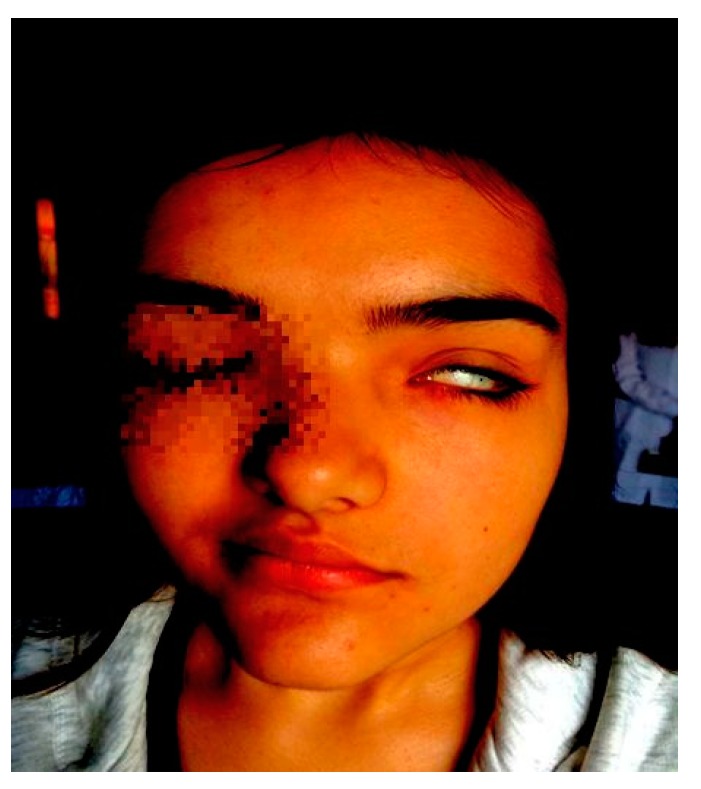
Right-sided deviation of the labial commissure, obliteration of the left nasolabial fold, and incomplete closure of the left eye (Patient 1).

**Figure 2 ijerph-16-01289-f002:**
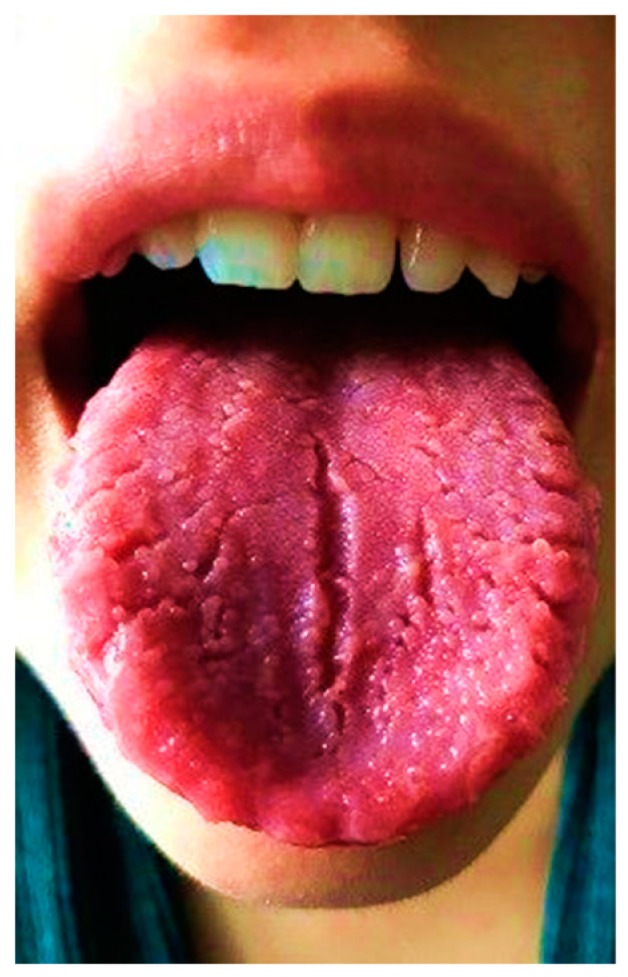
Fissured tongue (Patient 1).

**Figure 3 ijerph-16-01289-f003:**
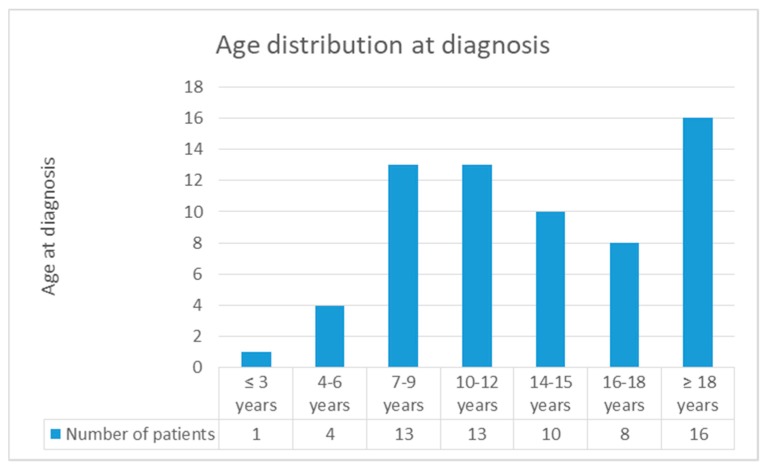
Age distribution at diagnosis.

**Figure 4 ijerph-16-01289-f004:**
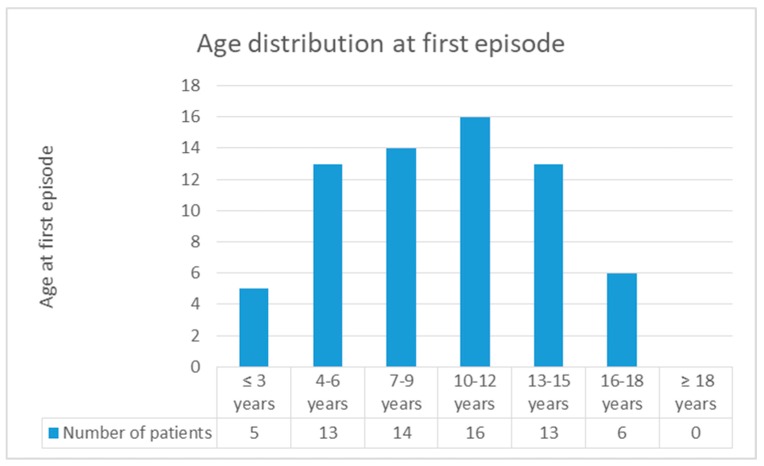
Age distribution at first episode.

**Table 1 ijerph-16-01289-t001:** Melkersson–Rosenthal syndrome in children and adolescents: literature review and case reports.

Reference	Sex *	Age **		Episodes ***	FP ^§^	Left FP	Right FP	Orofacial Oedema	Lingua Plicata	Treatment
		Diagnosis	Onset							
G.B. New, 1933 [12] ^¶^	F	31	4	5	Y	≥1	≥1	Y	NA	P
	M	25	12	3	Y	1	1	Y	NA	P
	M	19	2	≥2	Y	NA	≥1	Y	NA	P
K.A. Ekbom, 1950 [13]	F	25	16	6	Y	2	1	Y	Y	None
	NA	NA	8	1	Y	NA	NA	Y	Y	None
	M	20	15	4	Y	1	3	N	Y	None
Ehmann, 1962 [14]	NA	22 m	NA	NA	Y	NA	NA	Y	Y	NA
	NA	8	NA	NA	Y	NA	NA	Y	Y	NA
R.H. Kundstadter, 1965 [15]	M	5 y 6 m	2 y 6 m	4	Y	2	2	Y	N	C
G.A. Scott, 1968 [16]	M	8	6	2	Y	1	1	N	Y	None
	F	16	16	1	Y	0	1	N	Y	None
	F	15	NA	8	Y	≥1	≥1	Y	Y	None
	M	10	10	3	Y	1	2	Y	Y	None
	F	12	12	1	Y	0	1	N	Y	None
	M	62	10	>1	Y	>1	0	Y	Y	None
D. Dylewska, 1969 [17]	NA	9	NA	NA	NA	NA	NA	NA	NA	NA
P.K. Mukherjee, 1973 [18]	F	8	≤8	1	Y	1	1	Y	Y	None
AIT.J. Storrs, 1975 [17]	F	16	NA	NA	Y	NA	NA	Y	NA	C, H
H. Butenschon, 1976 [17]	M ^‡^	10–13								C, AB, ND
	F ^††^	5–14								
B. Roseman, 1978 [17]	F	7	NA	>1	Y	NA	NA	Y	Y	H
C. Lygidakis, 1979 [19]	F	12	7	3	Y	2	1	Y	N	None
M. May, 1981 [20] ^†^	M	NA	≤4	10	Y	1–9	1–9	Y	Y	NA
J. Neuhofer, 1984 [17]	F	15	NA	NA	Y	NA	NA	Y	Y	CF
V. Hridinova, 1985 [17]	NA	child	NA	NA	NA	NA	NA	NA	NA	NA
S. Yuzuk, 1985 [17]	F	13	NA	NA	N	0	0	Y	Y	P
K.M. Grundfast, 1990 [21]	NA	15	NA	3	Y	0 – 2	1 – 3	Y	Y	C
E. Truy, 1992 [17]	NA	child	NA	NA	Y	NA	NA	Y	Y	NA
L.T. Glickman, 1992 [22]	F	14	NA	NA	Y	NA	NA	N	Y	None
	M	15	NA	NA	NA	NA	NA	Y	NA	C
W.M. Zimmer, 1992 [23]	M	NA	6	NA	NA	NA	NA	NA	NA	C
	M	NA	0–10	NA	NA	NA	NA	NA	NA	
C. Bourgeois-Droin, 1993 [24]	F	10	10	2	N	0	0	Y	Y	C, CF
M. Pellegrino, 1993 [17]	M	13	NA	NA	Y	NA	NA	Y	Y	NA
C. Marques, 1994 [25] ^¶¶^	M	15	15	3	Y	NA	NA	Y	Y	C, CF
	NA	6	6	1	N	0	0	Y	Y	
H.A. Cohen, 1994 [10]	M	10	NA	>1	Y	0	1	Y	Y	NA
	F	10	NA	>1	Y	NA	NA	Y	Y	NA
	M	8	NA	>1	Y	NA	NA	Y	N	NA
	M	5	NA	NA	Y	1	0	Y	N	NA
E. Ruza Paz-Curbera, 1998 [17]	M	15	NA	NA	N	0	0	Y	N	C, H
S.L. Stein, 1999 [22]	F	12	12	NA	N	0	0	Y	Y	C, AB
	M	10	10	NA	N	0	0	Y	N	
P.E. Ziem, 2000 [17]	F	9	4	3	Y	3	0	Y	N	C, AB
D. Salgado Alves Vilela, 2002 [26]	M	17	≤16	4	Y	1	3	N	Y	C, ND
	F	23	14	6	Y	3	3	N	Y	
	F	16	4	3	Y	3	0	Y	Y	None
	F	20	2	3	Y	1	2	Y	Y	
H. Caksen, 2002 [10]	NA	child	NA	>1	Y	NA	NA	Y	Y	NA
I. Dodi, 2006 [27]	M	8	NA	NA	N	0	0	Y	N	C
H. Hentati, 2008 [28]	M	18	17/18	2	Y	1	0	Y	Y	C
H. Alp, 2009 [10]	M	10	NA	>1	Y	NA	NA	Y	N	NA
O. Ozgursoy, 2009 [7]	M	32	16	6	Y	2	3	Y	Y	C
	M	21	12	>1	N	0	0	Y	Y	C
S. Lourenaço, 2010 [29]	M	10	7	P	N	0	0	Y	Y	NA
R. Liu, 2013 [6]	M	45	6	>1	Y	>1	>1	N	Y	C
	M	16	4	NA	N	0	0	Y	Y	C
M.K. Elias, 2013 [30]	F	16	10	NA	Y	NA	NA	Y	Y	C
Ü. Yilmaz, 2014 [31]	NA	NA	13	2	Y	NA	NA	NA	Y	NA
Y. Lee, 2014 [10]	F	9	8	2	Y	0	2	Y	Y	C
S. Feng, 2014 [32] ^‡†^										
C.M. Rivera-Serrano, 2014 [2]	M	22	2	8	Y	4	4	Y	N	NA
	M	32	10	3	Y	1	2	Y	N	ND
	F	40	15	6	Y	0	6	Y	Y	NA
	F	31	7	6	Y	5	1	N	Y	ND
S. Bohra, 2015 [4]	F	16	16	1	Y	1	0	Y	Y	C, S
F. Kayhan, 2015 [3]	F	13	5	6	Y	0	6	Y	Y	C, S
G. Kayabasoglu, 2015 [33]	F	17	10	6	Y	3	3	Y	Y	C
A.G. Saini, 2016 [34]	F	8	2y6m	6	Y	3	3	Y	Y	C, ACV
S.S. Bakshi, 2016 [35]	F	9	7	2	Y	2	0	Y	Y	C
Z. Chu, 2016 [36]	M	12	10	>1	Y	0	1	Y	Y	C
L. Bordino, 2016 [37]	M	7	7	2	Y	1	1	Y	Y	NA
	F	11	9	2	Y	1	1	N	Y	C, ACV, S
J. Lalosevic, 2017 [5]	F	12	9	P	N	0	0	Y	N	C, AB
X.G. Xu, 2017 [38]	F	NA	12	2	Y	0	2	Y	Y	NA
C. Fantacci, 2018 [39]	F	8	5	2	Y	0	2	Y	Y	C, ACV, IVIG, S
G. Psillas, 2018 [31]	F	5	NA	N	Y	1	0	Y	Y	C
S. Savasta, 2018	F	14	11	2	Y	1	1	Y	Y	C, ACV, S
	F	7	3 y 1 m	4	Y	≥ 3	NA	N	Y	C, S
	F	9	18 m	2	Y	0	2	Y	NA	C, ACV, S

* Sex: F = Female, M = Male; ** age of the first episode and age of diagnosis; *** number of the episodes, including the first one; FP = Facial Palsy; Y = presence; N = absence; NA: data not available (when used for the age, NA refers to paediatric age); P = physical destruction of the involved mucocutaneous tissues (includes: boiling water local injections, external X-ray irradiation, surgical removal); C = corticosteroids; CF = clofazimine; H = antihistaminic drugs; AB = antibiotics; ND = surgical nerve decompression; S = symptomatic therapy (including analgesics, vitamins—B12 supplementation in particular—physiotherapy, and the prevention of complications, such as ocular infections with antibiotic eye drops); ACV = acyclovir; IVIG = intravenous immunoglobulins; ^¶^: Melkersson–Rosenthal’s symptoms described without a definite diagnosis (in 1933 the triad had not been described yet); ^†^ Other three cases reported in the article; ^‡^ four cases between 10 and 13 years; ^††^ 11 cases between 5 and 14 years; ^¶¶^ Other two cases reported in the article, both diagnosed before 10 years; ^‡†^ 18 Cases (8 M; 10 F) with first manifestation at under nine years old.

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
