# Peer review of "Melkersson–Rosenthal Syndrome in Childhood: Report of Three Paediatric Cases and a Review of the Literature"

_ijerph, 2019, doi:10.3390/ijerph16071289_

Round 1
Reviewer 1 Report
This manuscript focuses on pediatric presentation of MRS, adding 3 new case reports to those ones already reported and revising literature.
I wish to thank the Authors for giving me the opportunity to learn more about this rare disease, although I feel that some questions and concerns should be addressed:
line 14. Although it is well known that MRS is a rare condition, and that the prevalence of the disease in the pediatric age may be even more rare, “childhood MRS” has not been described so far as a specific disease
Line 57-76 Authors state (line 34-36) that only Miescher MRS might require mucocutaneous biopsy to be diagnosed, whilst MRS in oligosymptomatic or complete forms do not require any additional bioptic investigation as the diagnosis is a clinical one.
However, although 2.1 Patient 1 in their 2. Case report section showed evidence of a complete triad presentation, and the putative diagnosis of MRS was strengthened by familial history, tens of unhelpful (including screening for Gilbert disease !), confusing and potentially dangerous (e.g, CT scan and CSF examination) diagnostic test were performed.
This approach sounds as the opposite of precision medicine, which is considered at present the gold standard in approaching patients.
Anyway, if the Authors feel that they were right they should explain their diagnostic strategies step by step, from pretest hypothesis to interpretation of the results, and in which way single results prompted to further assessments.
Also, why patient received acyclovir without any evidence of viral infection? And why omeprazole for a short course of 5 days too?
Less relevant: Figures’ citations in the text are inverted, being that Fig 1 shows tongue and Fig 2 eye.
Once again:
- Lines 102-06. why a complete panel of lab test and autoimmunity screening when we had both facial palsy and fissured tongue as diagnostic criteria ?
- Lines 119-128 Why so many test and why acyclovir although viral checking tested negative ?
- Which need of vitamin supplementation in the 3 patients? And why different approaches?
Line 51-52 Why having “born from non consanguineous parents after uncomplicated pregnancy and natural delivery” should be relevant in this setting? If so, I would kindly ask Authors to give some examples of differential diagnoses with similar patterns (cheilitis, peripheral nerve palsy, fissured tongue) rising from children of consanguineous parents and/or after complicated pregnancy and/or cesarean delivery. The same for lines 85-86 and 117
Line 90. Was the patient previously treated with recombinant human erythropoietin (rHuEPO)? What about anti-EPO Abs assessment?
Lin 93: First recurrence of facial palsy in 2.2 Patient 2 seemed not related to MRS, being a consequence of hypertension-inuced , hemorrhagic stroke. Also, how was it possible that MRI performed on third episode (line 96) did not detect any radiologic consequence of the pfevious storke?
3. Methods, line 129-138
The choice to search for data from 1933 and including only “Melkersson-Roshental Syndrome” as a key word, thus ruling out oligo/monosymptomatic variants of a disease in which diagnostic criteria have not been established so far, does not help accuracy
4. Discussion
I feel that the discussion should be extensively rewritten if the purpose is, as it should be, to help readers to understand the main features of a (rare) disease and not only to provide a long list of findings without any guiding principle.
Author Response
1) line 14. Although it is well known that MRS is a rare condition, and that the prevalence of the disease in the pediatric age may be even more rare, “childhood MRS” has not been described so far as a specific disease
Indeed, “Childhood MRS” does not address a specific disease. The term has been replaced by “Melkersson – Rosenthal Syndrome (MRS) in children”, which is more appropriate (page 1 line 14).
2) Line 57-76 Authors state (line 34-36) that only Miescher MRS might require mucocutaneous biopsy to be diagnosed, whilst MRS in oligosymptomatic or complete forms do not require any additional bioptic investigation as the diagnosis is a clinical one.However, although 2.1 Patient 1 in their 2. Case report section showed evidence of a complete triad presentation, and the putative diagnosis of MRS was strengthened by familial history, tens of unhelpful (including screening for Gilbert disease !), confusing and potentially dangerous (e.g, CT scan and CSF examination) diagnostic test were performed.This approach sounds as the opposite of precision medicine, which is considered at present the gold standard in approaching patients.Anyway, if the Authors feel that they were right they should explain their diagnostic strategies step by step, from pretest hypothesis to interpretation of the results, and in which way single results prompted to further assessments.
As a good clinical practice standard, investigations were performed based on clinical history in order to exclude potentially dangerous differential diagnosis and eventually todetect any associated condition. Being a case series on retrospective studies, several investigations were performed in the course of the disease before diagnosis of MRS was made. Indeed, MRS is a rare condition, especially in children, and a clinical diagnosis needs to be supported by the exclusion of underlying conditions that are far more common in a pediatric setting. A brief section on differential diagnosis was added (see comments to reviewer#3), to help the readers understand its importance, and the rationale for ancillary investigations.A full list of exams which was performed in all three caseswas added inTable#1 as Supplementary Material.Cerebrospinal Fluid analysis were performed to rule out CNS viral infections in Patient #1 and Patient #3 and was avoided in Patient#2 considering her cerebrovascular disease history. Brain Computed Tomography was performed to rule out cerebrovascular diseases in Patient #1 and Patient #2and was avoided in Patient #3 (she had previously undergone MRI assessments). Biopsy was avoided in all three patients, given its invasiveness and limited usefulness, based on the current knowledge. Finally, the case reports were rewritten in order to clarify the need for further assessments in a step-by-step scheme (#3.Case Report, line 60-132).
3) Also, why patient received acyclovir without any evidence of viral infection?
Facial palsy was treated according to the 2013 American Academy of Otolaryngology – Head and Neck Surgery “Clinical Practice Guideline: Bell’s Palsy”. This guideline states that “Clinicians may offer oral antiviral therapy in addition to oral steroids within 72 hours of symptom onset for patients with Bell’s palsy. Option based on randomized controlled trials with minor limitations and observational studies with equilibrium of benefit and harm.” Patients were administered acyclovir until CSF or serologies resulted negative for herpetic infection. This concept was highlighted in the text (page10 line 289-290).
4) And why omeprazole for a short course of 5 days too?
Patient #1 received omeprazole for 5 weeks as a prophylaxis during the administration of corticosteroids. The “5 days” period previously reported was an erratum. Since this therapy is only relatively relevant to the description of MRS, it was omitted in the reviewed text.
5) Less relevant: Figures’ citations in the text are inverted, being that Fig 1 shows tongue and Fig 2 eye.
Figures and figures’ citations are now in the correct order (page 3 line 82-83; line 85).
6)Once again:
- Lines 102-06. why a complete panel of lab test and autoimmunity screening when we had both facial palsy and fissured tongue as diagnostic criteria ?
- Lines 119-128 Why so many test and why acyclovir although viral checking tested negative ?
Please, see answers to comment 1) and 2)
7) Which need of vitamin supplementation in the 3 patients?
Vitamin B supplementation was administered for its neurotrophic activity. More studies on the topic are needed to better understand its efficacy, but different studies have found it useful and unharmful for patients. Further bibliography is provided.
[Sun H, Yang T, Li Q, et al. Dexamethasone and vitamin B(12) synergistically promote peripheral nerve regeneration in rats by upregulating the expression of brain-derived neurotrophic factor. Arch Med Sci. 2012;8(5):924–930. doi:10.5114/aoms.2012.31623
Jalaludin MA, Methylcobalamin treatment of Bell's palsy.Methods Find Exp Clin Pharmacol. 1995 Oct;17(8):539-44.]
And why different approaches?
As we point out in the Discussion section (page 10 line 283), guidelines for treatment of MRS are not available, hence different therapeutic strategies have been used, in the course of the years, to achieve satisfactory results. The choice of intramuscular B12 supplementation was made to overcome compliance issues in adolescent patients.
8)Line 51-52 Why having “born from non consanguineous parents after uncomplicated pregnancy and natural delivery” should be relevant in this setting? If so, I would kindly ask Authors to give some examples of differential diagnoses with similar patterns (cheilitis, peripheral nerve palsy, fissured tongue) rising from children of consanguineous parents and/or after complicated pregnancy and/or cesarean delivery. The same for lines 85-86 and 117
Perinatal history is a common but necessary background information in pediatric case description. Consanguinity and family history can raise suspicion of genetic conditions (i.e. Moebius syndrome, CHARGE), whilst normal delivery rules out peripartal asfisxia and traumatic facial palsy which can eventually be misinterpreted in later life. We added a section regarding differential to the revised version of the manuscript (page 9-10 line 238-282)
9) Line 90. Was the patient previously treated with recombinant human erythropoietin (rHuEPO)? What about anti-EPO Abs assessment?
As we better explain in the revised version of the manuscript (page 3 line 93-97), the detection of anti-EPO antibodies is not routinely performed in a clinical setting. The autoimmune hypothesis has been postulated upon clinical evidence and “ex juvantibus” therapy response. The patient was administered rHuEPO: the persistence of low haemoglobin levels after this treatment, the increase of these levels after corticosteroid administration, and the inhibition of erythroid precursors growth after the addiction of the girl’s serum to the culture support the autoimmune hypothesis.
10) Line 93: First recurrence of facial palsy in 2.2 Patient 2 seemed not related to MRS, being a consequence of hypertension-inuced , hemorrhagic stroke.
We agree that the two events may just be coincidental. Stroke can cause contralateral central facial palsy when the upper motoneuronis affected. Besides, the symptoms completely resolved one month after their beginning. This was more clearly pointed out in the text (page 3 line 97-103). Even if potentially misleading, we felt it was important to clearly state this condition in the clinical history of the patient. Finally, all other events were consistent with an isolated peripheral facial palsy and, given the additional diagnostic criteria, a diagnosis of MRS could be made independently from this single unexpected event.
Also, how was it possible that MRI performed on third episode (line 96) did not detect any radiologic consequence of the previous stroke?
MRI showed non evolutive findings in the sense that no new lesions or abnormalities were evident. Gliotic evolution of the previous insult affecting the putamen was seen. In the revised paper, this concept has been better explained (page 3-4 line 105-106).
11)The choice to search for data from 1933 and including only “Melkersson-Roshental Syndrome” as a key word, thus ruling out oligo/monosymptomatic variants of a disease in which diagnostic criteria have not been established so far, does not help accuracy
The first clinical description of MRS dates back to 1928, when Ernst Gustaf Melkersson reported the possible association between orofacial swelling and facial palsy. Curt Rosenthal added the description of lingua plicata to the clinical picture in 1931. The first paediatric cases were described in 1933 and this information can be considered a result (not a method) of our research. Thus, the sentence “A literature search from the month of April 1933 to the month of August 2018” was replaced by the more accurate “We searched PubMed (Medline) through August 31, 2018”.Also, in this revised version, we indicated the complete search string used: [(“Melkersson-Rosenthal” AND “Syndrome”) OR “Miescher”] OR [“Cheilitis granulomatosa” AND “Miescher”] OR [“facial palsy” OR (“lingua plicata” OR “furrowed tongue” OR “scrotal tongue”) OR (“orofacial oedema” OR “facial swelling” OR “lip swelling”)] AND (“paediatric cases” OR “children”). (page 2 line 50-55).
12)I feel that the discussion should be extensively rewritten if the purpose is, as it should be, to help readers to understand the main features of a (rare) disease and not only to provide a long list of findings without any guiding principle.
We extensively re-organised the discussion section following your suggestions .
Reviewer 2 Report
The authors presented three cases of a rare disease, Childhood Melkersson – Rosenthal Syndrome (MRS), as well as provided an extensive literature review. The cases were carefully studied and the reports are of great value, especially for a rare disease such as MRS. A few minor comments:
Regarding diagnosis, discussion of biopsy would be welcome.
It would be great if the treatments in the literature could be summarized in Table 1.
Regarding Figure 2 was the consent from the patient obtained?
Page 9 Line 242: Cron should be Crohn.
Author Response
Regarding diagnosis, discussion of biopsy would be welcome.
As we report in the Introduction section, biopsy is mandatory to diagnose Merlkersson-Rosenthal Syndorme only in the case MRS in monosymptomatic. Therefore, since biopsy is an invasive testing, when the clinical presentation met the requirements needed for clinical diagnosis, biopsy was avoided. All the reported patients met the clinical criteria diagnosis. (page 9-10 line 270-272)
It would be great if the treatments in the literature could be summarized in Table 1.
We added a column to Table 1 (page 4-6 line 142-153) to summarize treatments
Regarding Figure 2 was the consent from the patient obtained?
Written informed consent for publication of clinical data and accompanying images was obtained from the parents of the patients, as stated in the manuscript. A copy of the consent form is available upon request to the Corresponding Author.(page 1-2 line 41-45)
Page 9 Line 242: Cron should be Crohn.
Correction made (page 10 line 272).
Reviewer 3 Report
A good paper, will significantly contribute to medical literature. Please answer the following-
Is there any reason why the search was restricted to 1993 and not prior?
These patients present to the ENT, neurological or dermatological specialty clinics, is there a possibility that there were missed cases from these clinics?
Do you have any comments on differential diagnoses for this condition?- angioedema, stroke, trigeminal neuralgia to name a few
Any comparative clinical presentation with adults or similarities? Any comments on the presentation in adults?
Do you have cases followed up beyond the age of 18, when patients become adults?
Any comments on the cranial nerves involved and description of the nerves involved apart from the facial nerve. There may or may not be a connection between combination of palsies of trigeminal and facial nerve. Some studies have suggested that the hypoglossal, glossopharyngeal, auditory and olfactory nerves may be implicated. Any comments on these
Author Response
1) Is there any reason why the search was restricted to 1993 and not prior?
The first clinical description of MRS dates back to 1928, when Ernst Gustaf Melkersson reported the possible association between orofacial swelling and facial palsy. Curt Rosenthal added the description of lingua plicata to the clinical picture in 1931. The first paediatric cases were described in 1933 and this information can be considered a result (not a method) of our research. Thus, the sentence “A literature search from the month of April 1933 to the month of August 2018” was replaced by the more appropriate “We searched PubMed (Medline) through August 31, 2018”. (page 2 line 50-52)
2) These patients present to the ENT, neurological or dermatological specialty clinics, is there a possibility that there were missed cases from these clinics?
This is one the major problems of this syndrome. As we point out in the Discussion section, MRS is both mis- and under-diagnosed. The aim of this article is to raise awareness on such a rare syndrome. (page 4 line 134-136)
3) Do you have any comments on differential diagnoses for this condition?- angioedema, stroke, trigeminal neuralgia to name a few
We added a section regarding differential to the revised version of the manuscript (page 9-10 line 238-282)
4) Any comparative clinical presentation with adults or similarities? Any comments on the presentation in adults?
Children and adults have the same clinical presentation and prognosis. Prognosis is mostly influenced by the rate of recurrences. A brief deepening on this comparation was added to the Discussion section.
5) Do you have cases followed up beyond the age of 18, when patients become adults?
The three patients we have followed have not turned 18 yet. We unfortunately have no direct experience on adult MRS.
6) Any comments on the cranial nerves involved and description of the nerves involved apart from the facial nerve. There may or may not be a connection between combination of palsies of trigeminal and facial nerve. Some studies have suggested that the hypoglossal, glossopharyngeal, auditory and olfactory nerves may be implicated. Any comments on these
We added a section regarding cranial nerves apart from the facial nerve in the revised version of the manuscript (page 9-10 line 238-282)
Round 2
Reviewer 1 Report
No further comments